# Designing and evaluation of the effect of community-based intervention on breast self-examination among reproductive-aged women in Ethiopia: A Cluster Randomized Controlled Trial

Feleke Doyore Agide [1,2]*, Gholamreza Garmaroudi[3]*, Roya Sadeghi[3], Elham Shakibazadeh[3], Mehdi Yaseri[4], Zewdie Birhanu[5]

**1** Department of Public Health, College of Medicine and Health Sciences, Wachemo University, Hossana, Ethiopia, **2** Department of Health Education and Promotion, School of Public Health, International Campus, Tehran University of Medical Sciences, Tehran, Iran, **3** Department of Health Education and Promotion, School of Public Health, Tehran University of Medical Sciences, Tehran, Iran, **4** Department of Epidemiology and Biostatistics, School of Public Health, Tehran University of Medical Sciences, Tehran, Iran, **5** Department of Health, Behavior and Society, Institute of Health Sciences, Jimma University, Jimma, Ethiopia

* feledoag@yahoo.com (FDA); garmaroudi@tums.ac.ir (GG)

## Abstract

### Background

Though early intervention saves many lives worldwide, breast cancer remains a leading cause of cancer among women in Ethiopia. This study, therefore, aimed to evaluate community-based interventions promoting breast self-examination using the Health Belief Model.

### Methods

A cluster randomized controlled trial followed by a cross-sectional study lasting six months was used to evaluate the effectiveness of the community-based educational intervention on breast self-examination among reproductive-aged women in Ethiopia. A total of 810 participants were randomly assigned in a 1:1 ratio and assessed at baseline, three months, and six months post-intervention. A general linear model for repeated measures was used to examine the mean differences in study variables across time points. Non-parametric tests (Cochran's Q) were employed to analyze dichotomous variables related to breast self-examination practices. Path analysis was conducted to examine the interactions among the constructs of the Health Belief Model.

### Results

A total of 810 reproductive-aged women participated in the study, yielding a 100% response rate at baseline. The mean age of participants was 33.2±7.7 years in the

**Data availability statement:** All relevant data are within the manuscript and its Supporting information files. We confirm that the minimal data set is already included and, where applicable, additional files have been uploaded to the File Inventory as Supporting information files.

**Funding:** The author(s) received no specific funding for this work.

**Competing interests:** The authors have declared that no competing interests exist.

intervention group and 33.5±8.1 years in the control group. The proportion of women performing breast self-examinations increased from 33.3% at baseline to 59.9% at the end of the intervention. And the Comprehensive knowledge about breast self-examination rose from 11.7% to 69.1% over the same period. Perceived susceptibility, perceived severity, knowledge, and health motivations had a statistically significant mean difference between the intervention and control groups (p < 0.0001). We registered PACTR database (https://pactr.samrc.ac.za/): "PACTR201802002902886".

## Conclusions

The study found that there is a strong interplay between the likelihood of performing breast self-examination and perceived susceptibility, perceived severity, knowledge, and health motivations. Field specialists should figure out the problem related to perception and awareness through intensive health promotion interventions.

## Trial registration

Registered in the Pan African Clinical Trial Registry (www.pactr.org) database, and the unique identification number for the registry is **PACTR201802002902886.**

## Introduction

Breast cancer (BC) is a non-communicable disease characterized by the uncontrolled growth of cells, most commonly originating in the milk ducts (ductal carcinoma), which carry milk from the breast to the nipple [1–3]. According to global estimates from 2020, breast cancer affects approximately 2.3 million women each year and is the leading cause of cancer-related deaths among women in both developed and developing countries [1,2]. While its prevalence remains higher in developed nations, the incidence in developing countries, including Ethiopia, is rising at an alarming rate [4]. In various research reports, risk factors for breast cancer related to lifestyle, such as eating unhealthily, being obese, and using toxic substances, are linked to higher morbidity and mortality rates [4,5]. Furthermore, factors such as increased life expectancy, urbanization, and the adoption of Western lifestyles have further accelerated the rise in incidence in low- and middle-income countries [6,7].

The timing of detection determines the effectiveness of breast cancer treatment [8,9]. Ethiopian women typically present for care at a late stage in the disease, where treatment is most ineffective, and while system-related barriers to care account for a portion of that delay in access, a lack of awareness of breast cancer symptoms accounts for a stalled initiation of action [8]. For this reason, early breast screening through breast self-examination is the preferred method for early detection of breast cancer. Previous literature documented breast self-examination, and early detection has shown improvement in the treatment outcome of breast cancer [7,8].

In Ethiopia, despite various breast cancer prevention mechanisms suggested by health professionals, early recognition of the symptoms and self-referral for treatment

are still in question, and their chances of survival are nil due to a late report [4,8–10]. Breast self-examination has a significant impact on early diagnosis and prompt care [10]. Several observational studies have been conducted on breast self-examination among women, but none of these were interventional studies among women of reproductive age in Ethiopia [11–13]. Thus, health promotion interventions have immense potential for awareness creation, early detection, and improved survival [3,14,15].

Therefore, the aim of this study was to design and evaluate community-based interventions on breast self-examination that used the CONSORT flow diagram of a cluster randomized control trial of two groups (Fig 1) in the theoretical framework of the Health Belief Model (HBM). HBM is a socio-psychological model that attempts to explain and predict health behaviors in terms of certain belief patterns by focusing on the attitudes and beliefs of individuals. It was developed by social psychologists to explain the lack of public participation in health screening and prevention programs. Since then, it has been adapted to a variety of long- and short-term health behaviors, including breast screening behaviors [16]. The

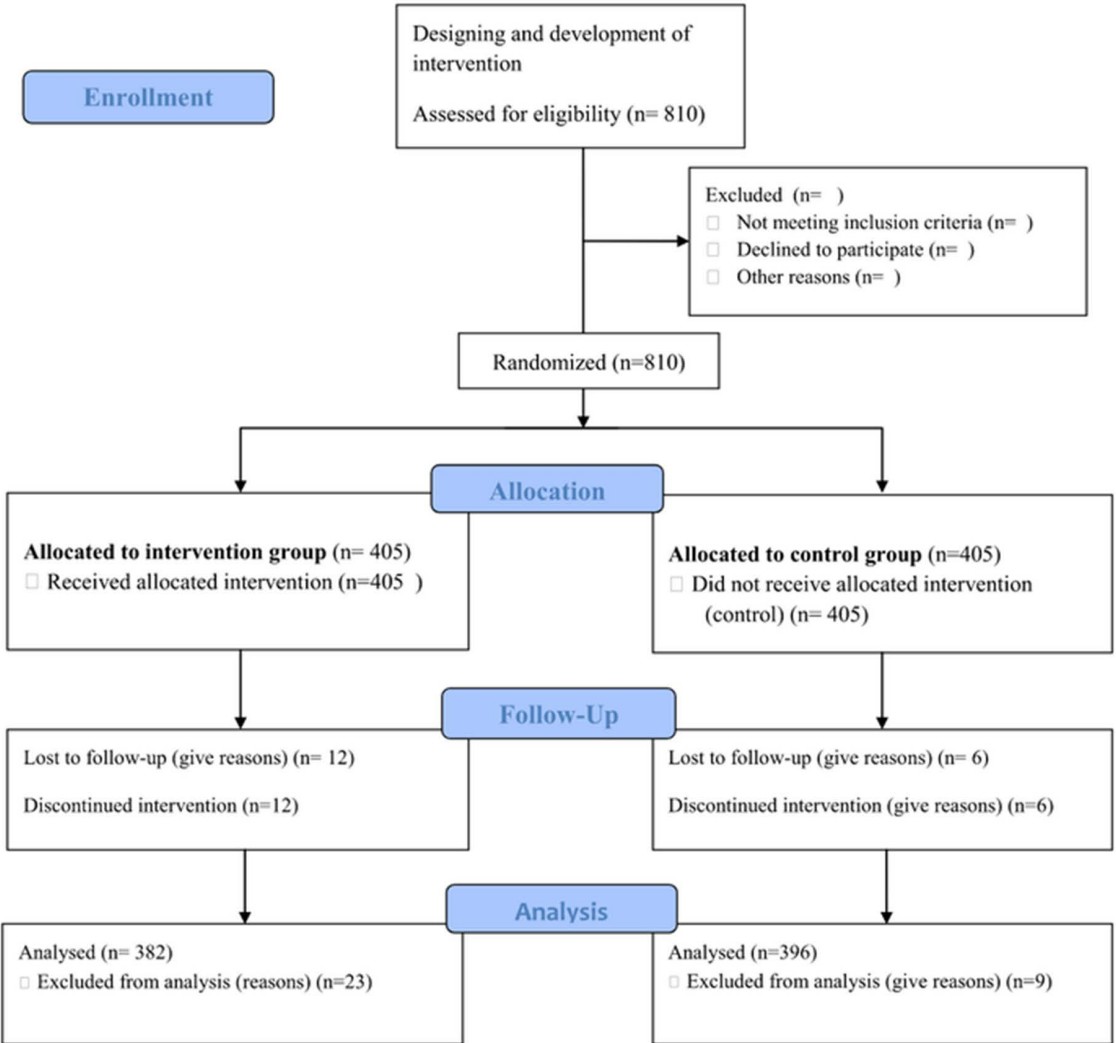

**Fig 1. CONSORT flow diagram of the progress through the phases of a parallel randomized trial of two groups (that is, enrolment, intervention allocation, follow-up, and data analysis).** Available at: http//www.consort-statement/flow-diagram.

HBM addresses the individual's perceptions of the threat posed by a health problem (susceptibility, severity), the benefits of avoiding the threat, and factors influencing the decision to act (barriers, cues to action, and self-efficacy); it also states specific health beliefs related to the health problem and recommended health actions that influence the likelihood of taking recommended health actions (breast self-examination) [14,16–19] (Fig 2).

## Materials and methods

### Study setting

This study was conducted on women of reproductive age in the Hadiya zone of Southern Ethiopia. There are 10 rural districts, two city administrations, and 332 kebeles and Kifleketemas within the zone. Hossana, its capital city, is 230 kilometers from Addis Ababa. The zone's population is estimated to be 1,850,104 people. Females of childbearing age (15–49) are estimated to number 193,967. The overall number of health institutions by type in the zone corresponds to the number of kebeles and more. Health extension workers and community health agents play an important role in the prevention of communicable and non-communicable diseases. The research was carried out between April 2018 and April 2019.

### Study design and populations

A cluster randomized controlled trial followed by a cross-sectional study lasting six months was used to evaluate the effectiveness of the community-based educational intervention on breast self-examination among reproductive-aged women in Ethiopia. A total of 810 participants were randomly assigned in a 1:1 ratio to either the intervention or control group and were evaluated at baseline, three months, and six months post-intervention. The intervention group received a specially designed educational program, while the control group received the usual services provided by health extension

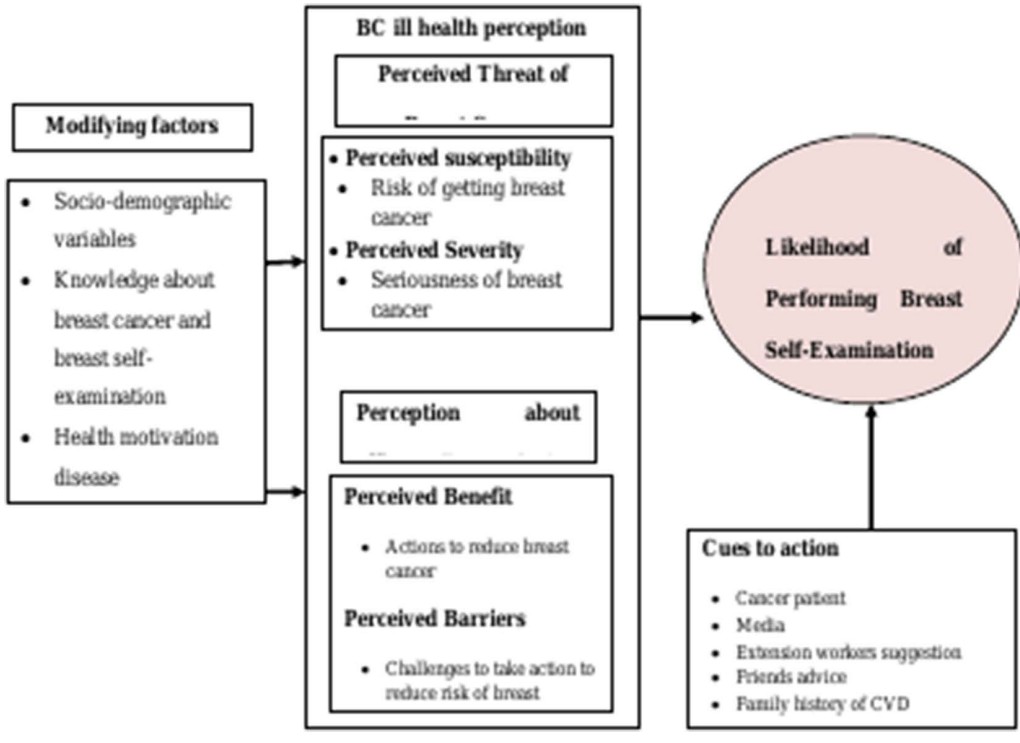

**Fig 2. Conceptual framework of the research [16,22].**

workers. The study included women in the reproductive age group (15–49) who were physically and mentally competent to provide informed consent, could follow the intervention provided without assistance, and were ready to provide data to the researcher. Exclusion criteria comprised individuals who were unable to remain in the study until completion or who moved away during the intervention period. Specifically, participants who left the study area before completing all three data collection points were excluded, as were those who attended fewer than two of the specially prepared training sessions. For the qualitative part, at the beginning, an exploratory qualitative study with women and health workers was conducted and published elsewhere, utilizing focus group discussions (FGDs) and in-depth interviews, respectively.

## Recruitment, sampling and sample size determination

Hadiya Zone consists of twelve districts, including ten rural districts and two city administrations at the time of data collection. For sampling, six districts were randomly selected. From these selected districts, 30 kebeles were chosen and evenly divided into intervention and control groups, with 15 kebeles in each group. To select participants within each kebele, systematic sampling was used by selecting every $K^{th}$ household. To minimize information contamination between groups, random assignment ensured that control and intervention sites were geographically distant. Since this was a health-promotion intervention, intervention and control woreda (districts) were purposely selected from widely separated locations. All kebeles within the intervention districts were assigned to the intervention group, and all kebeles within the control districts were assigned to the control group. Importantly, randomization was conducted at the district level rather than individually.

A total of 810 participants were proportionally allocated across the selected kebeles, following the WHO sampling recommendations [20]. For participant enrollment, eligible women were listed and invited to participate through oral invitations made by the principal investigator and health extension workers in each kebele. Upon agreeing to participate, appointments were scheduled, eligibility was confirmed, baseline data were collected, and participants were assigned to either the intervention or control group. Oral informed consent was obtained from the women who agreed to participate in the study. Participants in the randomized controlled trial were assigned 1:1 to the intervention and control arms under a cluster randomized controlled trial after informed consent and the collection of baseline data. The different kebeles were alphabetically coded (A, B, C, D, and so on), and participants attached to each kebele were given numerical codes (for example, participants in kebele A were numbered 1001, 1002, 1003, and so on). Interventionists, data collectors, and statisticians were not the same people. The baseline data collection recruitment began on April 1, 2018. Then, on May 7, 2018, the first education intervention was delivered and continued for three months, every 15 days. The follow-up data was collected from both groups after three months and at the end of six months.

Regarding sample size determination, since this was a cluster randomized controlled trial, the sample size was calculated using the double population proportion formula. It was based on 77.6% of participants having knowledge about breast cancer and breast self-examination ($P_1 = 77.6\%$) [21], and an assumed 10% increase in screening prevalence in the intervention group ($P_2 = 87.6\%$). The coefficient of variation (k) between districts was assumed to be 0.25 due to lack of prior estimates. With a 5% margin of error, 5% significance level (two-sided), and a 95% confidence interval, the initial calculated sample size was 368. To account for non-response and recording errors, the sample size was increased by 10% to 405. Considering a design effect of 2, the final sample size was set at 810 participants.

## Measurement and variables

The likelihood of performing breast self-examination is the intended outcome of this study (perceived benefits minus perceived barriers). The exposure variables were socio-demographic factors, knowledge of breast cancer and its screening methods, and past behaviors related to screening. Socio-demographic characteristics include age, marital status, religion, place of residence, educational and occupation status, and with whom the respondents are currently living. There are knowledge questions with a response format of 'yes' or 'no'. Respondents were asked not to guess but to mark the 'I don't know'

answer possibility if they did not know the correct answer. Knowledgeable are those respondents who answered above the mean values all the knowledge questions about breast cancer and its screening methods. Not knowledgeable were those respondents who could answer below above the mean values all the knowledge questions about breast cancer and its screening methods. Perceived susceptibility is the respondent's self-perception of vulnerability to breast cancer, measured by the summed score of related belief items on a 5-point Likert scale. Perceived severity of breast cancer is the respondent's held belief concerning the effects of breast cancer's seriousness, measured by the summed score of related belief items on a 5-point Likert scale. Perceived benefits of breast self-examination are the respondent's beliefs about the effectiveness of the method as a strategy for breast cancer prevention, measured by the summed score of related belief items on a 5-point Likert scale. Perceived barriers to performing breast self-examination are the respondent's beliefs about the ease of performing the given preventive action. Self-efficacy is the respondent's self-confidence to perform breast self-examination by herself in any condition and elsewhere to prevent breast cancer, as measured by the summed score of related belief items on a 5-point Likert scale. Negatively worded items were reversed before calculating a summed score for each concept. Cues to actions are conditions that might facilitate them to perform breast self-examination in the respondents' surroundings with a response format of 'yes or no'. Past behavior (practice) is the experience of the reproductive-aged women undergoing breast self-examination at least once in the recommended period to prevent breast cancer, measured with nominal measurements. We used Cronbanch's alpha for reliability analysis of the questionnaire. All the Chronbach's alpha coefficients were greater than 0.7. Accordingly, the alpha of knowledge of breast cancer and breast self-examination $\alpha = 0.91$, perceived susceptibility to breast cancer $\alpha = 0.79$, perceived severity of breast cancer $\alpha = 0.82$, perceived benefits of breast self-examination $\alpha = 0.86$, perceived barrier of breast self-examination $\alpha = 0.81$, perceived self-efficacy of breast self-examination $\alpha = 0.81$, and cue to action to breast cancer and breast self-examination $\alpha = 0.73$.

## Procedures (Intervention Description)

An educational intervention was prepared based on Health Belief Model constructs, which are interlinked with breast self-examination. The intervention emerged from qualitative components that were treated as distal factors in the model and used as a key basis for the intervention design. Since the study was guided by the Health Belief Model, the content of the sessions was described based on the model's constructs, comprehensive knowledge, and the enhancement of health motivation. Educational intervention on breast self-examination was delivered through training and teaching using various methods and materials, such as posters. Regarding the process, enumerated lists of participants were obtained from the registry books of health extension workers after obtaining consent. Immediately after baseline data collection, participants were categorized into intervention and control groups. All kebeles within intervention districts were categorized as intervention groups, and those within control districts as control groups. Since health promotion interventions are prone to contamination, random assignment was done at the district level rather than at the individual level. All required information from both groups was collected, recorded on a temporarily prepared attendance sheet, and followed accordingly. Participants in the intervention arm received a community-based educational intervention every 15 days for three months and registered their names and phone numbers (including family phone numbers) for tracking and reminder purposes. All participants were assured of confidentiality throughout the process.

The participants were given an appointment at a health center or health post near them where the usual community forum was being conducted. The interventionists, together with health extension workers, delivered community-based interventions. We used health extension workers from different (not included) woreda to deliver specially prepared health education or intervention for the intervention group; whereas, we used the existing health extension workers for the control group for usual education. These participants only received a welcome message at the beginning to validate their entry into the study and a message at the end of the follow-up to thank them for their participation. However, at the end of the six months, the same education was provided for the control groups. However, no data were collected after this intervention. This was done to overcome ethical issues in randomized trials for the benefit of control groups.

## Data collection and quality management

Data were collected using pre-tested, interviewer-administered structured questionnaires. Questionnaires were translated into the local language and then translated back to English by another person to maintain consistency. Two days of training were given for data collectors and supervisors. Supervisors and the principal investigator performed immediate supervision on a daily basis.

## Data processing and analysis

The data were analyzed by SPSS V. 24.0. Prior to analysis, the data were checked for normality and homogeneity. The Kolmogorov-Smirnov test was used to test the null hypothesis that a set of data comes from a normal distribution ($P > 0.05$). Intervention outcomes were analyzed following the reporting standards of the consolidated standard of reporting trials (CONSORT standards). For intervention and control arms comparison, a general linear regression model for repeated measures was used to assess the effectiveness of the intervention and predict independent predictors of self-examination. Non-parametric tests (Cochran Q) were used for dichotomous variables to measure the effect size of the intervention on breast self-examination. Path analysis was used to determine the direct and indirect effects of variables and to estimate the values of coefficients in the underpinning linear model at the end of six months.

## Ethics

Ethical approval for this study was obtained from the Research Ethical Review and Approval Board (RERB) of Tehran University of Medical Sciences, International Campus (TUMS-IC) under the reference number IR.TUMS.SPH. REC.1396.4088, as well as from the Research and Ethical Review Committee (RERC) of the Southern Ethiopia Regional Health Bureau (Reference No. 6-19/5524). The study was conducted in full compliance with the ethical principles outlined in the Declaration of Helsinki (Association, 2009). Official letters from both TUMS-IC and the Southern Ethiopia Regional Health Bureau were submitted to the Hadiya Zone Health Department to obtain legal permission to carry out the study. Informed verbal consent was obtained from all participants after a thorough explanation of the study's objectives and potential benefits. Participants were informed of their right to withdraw from the study at any time without any consequences. Confidentiality was assured throughout the study; participants were informed that their responses would remain confidential and that their names would not be recorded or mentioned in any reports. To address ethical concerns commonly associated with randomized trials, the same educational intervention provided to the intervention group was also offered to the control group at the end of the data collection period. This ensured that the control group also benefited from the study. Finally, the study was registered with the Pan African Clinical Trial Registry (www.pactr.org), and the unique identification number for the registry is PACTR201802002902886.

## Results

### Baseline and socio-demographic characteristics of the participants

Eight hundred and ten reproductive-aged women participated in the study, giving a response rate of 100% at baseline. Baseline data were collected from both the intervention and control groups before the start of the educational intervention. Follow-up data were then collected at three and six months after the intervention to assess its effect. Table 1 shows the socio-demographic characteristics of the participants at baseline. Accordingly, in the baseline survey, the mean age of the participants was 33.2 (SD±7.7) and 33.5 (SD±8.1) years in the intervention and control arms, respectively. Accordingly, a total of 810, 792 and 778 reproductive-aged women participated at baseline, three, and six months of the study, respectively (Table 1).

### Knowledge about breast cancer and breast self-examination

Fig 3 illustrates the participants' knowledge about breast cancer and breast self-examination in both the intervention and control groups over the study period. In the intervention group, the mean score for comprehensive knowledge significantly

**Table 1. Socio-demographic characteristics of the participants at baseline.**

| Variables | Category | Intervention and Control Categories (Measurement Time) | | | |
|---|---|---|---|---|---|
| | | Baseline | | | |
| | | Intervention group (n = 405) | | Intervention group (n = 405) | |
| | | No. | % | No. | % |
| Age | 15-24 | 80 | 19.8 | 92 | 22.7 |
| | 25-34 | 166 | 41.0 | 141 | 34.8 |
| | 35-44 | 136 | 33.6 | 129 | 31.9 |
| | 45-49 | 23 | 5.7 | 43 | 10.6 |
| Current Residence | Rural | 205 | 50.6 | 320 | 79.0 |
| | Urban | 200 | 49.4 | 85 | 21.0 |
| Religion | Protestant | 289 | 71.4 | 308 | 76.0 |
| | Orthodox | 70 | 17.3 | 71 | 17.5 |
| | Muslim | 31 | 7.7 | 14 | 3.5 |
| | Catholic | 15 | 3.7 | 12 | 3.0 |
| Marital status | Single | 32 | 7.9 | 45 | 11.1 |
| | Married | 350 | 86.4 | 344 | 84.9 |
| | Divorced | 23 | 5.7 | 16 | 4.0 |
| Educational Status | Can't read and write | 265 | 65.4 | 188 | 46.4 |
| | Can read and write | 95 | 23.5 | 102 | 25.2 |
| | Primary school | 11 | 2.7 | 30 | 7.4 |
| | High school | 14 | 3.5 | 45 | 11.1 |
| | College and above | 20 | 4.9 | 40 | 9.9 |
| Occupational status | House wife | 282 | 69.6 | 285 | 70.4 |
| | Employee | 36 | 8.9 | 28 | 6.9 |
| | Merchant | 28 | 6.9 | 39 | 9.6 |
| | Private business | 33 | 8.1 | 32 | 7.9 |
| | Students | 26 | 6.4 | 21 | 5.2 |
| Categorized Income | <= 500 | 282 | 69.6 | 268 | 66.2 |
| | 501-1000 | 75 | 18.5 | 87 | 21.5 |
| | 1001-1500 | 9 | 2.2 | 22 | 5.4 |
| | > 1500 | 39 | 9.6 | 28 | 6.9 |

increased from 4.0 ± 1.82 at baseline to 13.0 ± 1.62 at three months, and remained consistent at 13.0 ± 1.79 at six months post-intervention. However, in the control group, the mean (4.0 ± 2.11) showed a slight increase after three months and at the end of the intervention, whereas there was no significant difference between the two times of data collection in the control group (6 ± 1.75 and 6 ± 1.78) (Fig 3).

## Past behaviors (practice) related to breast self-examination and its screening procedure

Table 2 presents participants' past behavior regarding breast self-examination and their previous screening practices. At baseline, nearly all participants (95.6%) reported having heard of breast cancer, with 94.6% in the intervention group and 97.0% in the control group. Regarding screening methods, 36.9% (299/810) of participants had heard of breast self-examination as a method at baseline. However, after intervention, in the intervention group, all the participants heard of breast self-examination as a method. The control group showed a slight difference in hearing of breast self-examination from baseline to endline as depicted in Table 2 below (Table 2).

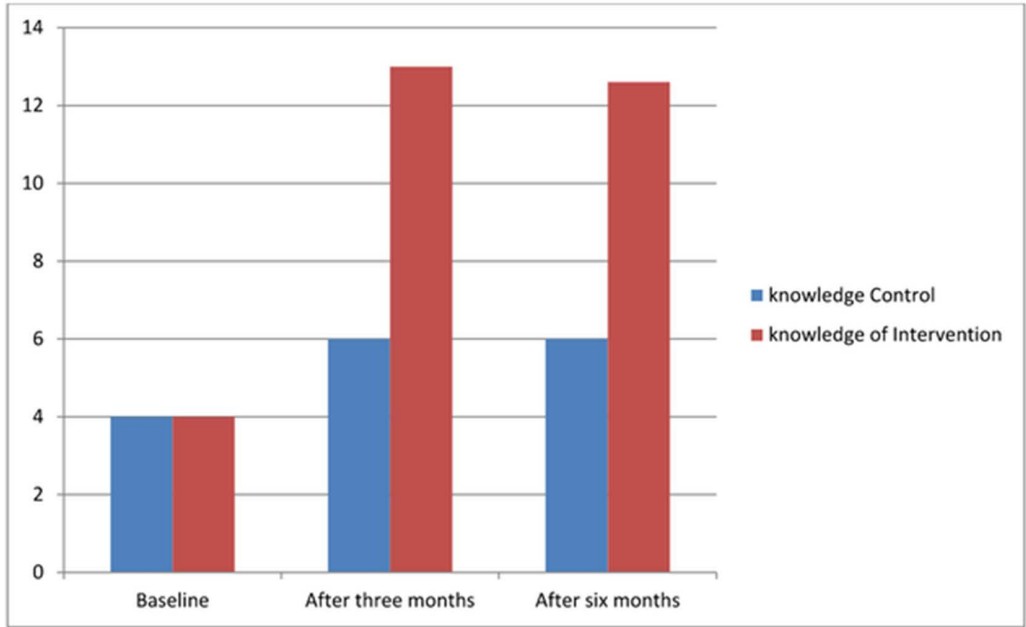

**Fig 3. Knowledge of the participants about breast cancer and breast self-examination in intervention and control groups across the study period.**

## Perception towards breast cancer and breast self-examination

Table 3 summarizes the participants' perception scores related to breast cancer and breast self-examination. At baseline, 33.3% of participants reported a likelihood of performing breast self-examination, while only 11.7% demonstrated comprehensive knowledge. By the end of the intervention, these figures had increased to 59.9% and 69.1%, respectively.

Likewise, the perception of threat appraisals (i.e., perceived susceptibility to and perceived severity of breast cancer) had respective mean scores of (mean±standard deviation) in the intervention ($17.6\pm4.5$ and $39.7\pm8.9$) and in the control group ($17.2\pm4.8$ and $38.7\pm9.5$) at baseline. However, there was a significant mean difference after intervention and at the end of six months ($p < 0.0001$). As far as the likelihood of performing breast self-examination is concerned, the perceived benefits and barriers at baseline had mean scores of $53\pm10.0$ and $71\pm10.4$ in the intervention group, and $53\pm9.8$ and $68\pm10.9$ in the control group, respectively. After three months and again at six months post-intervention, the intervention group showed a significant improvement in these scores, reaching $60\pm4.29$ and $60\pm4.35$, respectively, with a statistically significant difference compared to the control group ($p < 0.0001$) (Table 3).

## The independent predictors of likelihood of performing breast self-examination (Regression analysis)

A general linear model of repeated measures was used to assess the effect of interventions on the variables of the study. Table 4 presents the General Linear Regression Model analysis of repeated measures to compare the mean difference (two-way repeated measures of ANOVA). Accordingly, health belief model constructs, knowledge, and health motivations had a statistically significant mean difference between intervention and control groups ($p < 0.001$). Likewise, the mean score of perceived barriers was statistically significantly reduced in the intervention group after three and six months (mean difference = −3.697 between time 1 and time 2 and −3.495 between time 2 and time 3). However, the mean difference in cues to action was less than one, indicating that this construct showed the least variance explained by the intervention in the given context (Table 4).

**Table 2. Descriptive frequencies of past behaviors related to breast cancer and its screening among reproductive aged women in Ethiopia.**

| Variables | | Categories | Baseline Intervention group No | % | Control group No | % | After three months Intervention group No | % | Control group No. | % | After six months Intervention group No. | % | Control group No. | % | p-value |
|---|---|---|---|---|---|---|---|---|---|---|---|---|---|---|---|
| Ever heard breast cancer before? | | Yes | 383 | 94.6 | 391 | 97.0 | 393 | 100.0 | 394 | 98.7 | 382 | 100.0 | 391 | 98.7 | 0.006 |
| | | No | 22 | 5.4 | 14 | 3.0 | 0 | 0.0 | 5 | 1.3 | 0 | 0.0 | 5 | 1.3 | |
| Ever heard screening methods? | | Yes | 161 | 39.8 | 138 | 34.1 | 393 | 100.0 | 142 | 35.3 | 382 | 100.0 | 140 | 35.4 | <0.001 |
| | | No | 244 | 60.2 | 267 | 65.9 | 0 | 0.0 | 258 | 64.7 | 0 | 0.0 | 256 | 64.6 | |
| Source of information | Health Worker | Yes | 157 | 97.5 | 119 | 86.2 | 393 | 100.0 | 124 | 87.7 | 382 | 100.0 | 122 | 87.1 | <0.001 |
| | Media | Yes | 99 | 61.5 | 84 | 60.9 | 177 | 45.0 | 72 | 50.7 | 173 | 45.3 | 71 | 50.7 | <0.001 |
| | Relative | Yes | 99 | 61.5 | 84 | 60.9 | 139 | 35.4 | 91 | 64.1 | 135 | 35.3 | 89 | 63.6 | <0.001 |
| | Friends | Yes | 81 | 50.3 | 59 | 42.8 | 167 | 42.5 | 68 | 47.9 | 164 | 42.9 | 67 | 47.9 | 0.423 |
| Ever screened your breast before? | | Yes | 47 | 29.2 | 23 | 17.2 | 107 | 27.2 | 32 | 22.5 | 99 | 25.9 | 22 | 15.7 | <0.001 |
| Time of screening | | 2 Months Before | 5 | 10.6 | 2 | 8.7 | 7 | 6.5 | 2 | 6.3 | 7 | 7.1 | 2 | 8.7 | <0.001 |
| | | 6 Months Before | 29 | 61.7 | 9 | 39.1 | 64 | 59.8 | 9 | 28.1 | 60 | 60.6 | 5 | 21.7 | |
| | | A year ago | 13 | 27.7 | 9 | 39.1 | 36 | 33.6 | 19 | 59.4 | 32 | 32.3 | 14 | 60.9 | |
| | | 2 years ago | 0 | 0.0 | 3 | 13.0 | 0 | 0.0 | 2 | 6.3 | 0 | 0.0 | 2 | 8.7 | |
| Method of screening used? | | BSE | 32 | 68.1 | 15 | 65.2 | 95 | 88.8 | 25 | 78.1 | 90 | 90.9 | 18 | 78.3 | <0.001 |
| | | BCE | 12 | 25.5 | 5 | 21.7 | 9 | 8.4 | 5 | 15.6 | 9 | 9.1 | 5 | 21.7 | |
| | | Mammography | 3 | 6.4 | 3 | 13.0 | 3 | 2.8 | 2 | 6.3 | 0 | 0.0 | 0 | 0.0 | |
| Frequency of breast screening | | Sometimes | 27 | 57.4 | 13 | 56.5 | 65 | 60.7 | 19 | 59.4 | 57 | 57.6 | 12 | 52.2 | <0.001 |
| | | Usually | 1 | 2.1 | 2 | 8.7 | 5 | 4.7 | 2 | 6.3 | 5 | 5.1 | 2 | 8.7 | |
| | | Consistently | 4 | 8.5 | 3 | 13.0 | 16 | 15.0 | 7 | 21.9 | 16 | 16.2 | 6 | 26.1 | |
| | | Others** | 15 | 31.9 | 5 | 21.7 | 21 | 19.6 | 4 | 12.5 | 21 | 21.2 | 3 | 13.0 | |

*Based on descriptive statistics; b. Adjustment for multiple comparisons.**one in a time, when they were ill.

**Table 3. Mean scores of perception, health motivations and knowledge of the participants about breast cancer and its screening at baseline, after three months and after six months of intervention.**

| Variables | Score ranges | Baseline Intervention Mean±SD | Control Mean±SD | p-value | After three months Intervention Mean±SD | Control Mean±SD | p-value | After six months Intervention Mean±SD | Control Mean±SD | p-value |
|---|---|---|---|---|---|---|---|---|---|---|
| Perceived Susceptibility | 5-25 | 17.6±4.5 | 17.2±4.8 | 0.062 | 20.7±2.2 | 17.1±4.8 | <0.001 | 20.8±2.1 | 17.0±4.8 | <0.0001 |
| Perceived Severity | 11-55 | 39.7±8.9 | 38.7±9.5 | 0.078 | 50.0±2.6 | 43.2±7.7 | <0.001 | 50.2±2.5 | 43.5±7.5 | <0.0001 |
| Perceived Benefits | 14-70 | 53.0±10.0 | 53.0±9.8 | 0.057 | 60.0±4.3 | 54.0±9.6 | <0.001 | 60.0±4.4 | 54.0±9.5 | <0.0001 |
| Perceived Barrier | 19-95 | 71.0±10.4 | 68.0±10.9 | 0.047 | 81.0±4.4 | 67.0±11.6 | <0.001 | 81.0±4.3 | 67.0±11.6 | <0.0001 |
| Self-efficacy | 13-65 | 47.0±9.5 | 46.5±10.5 | <0.001 | 55.6±3.8 | 46.8±10.3 | <0.001 | 55.6±3.6 | 46.8±10.3 | <0.0001 |
| Cues to action | 0-7 | 3.5±2.0 | 4.0±1.9 | 0.002 | 4.6±1.9 | 4.02±1.9 | <0.001 | 4.6±1.9 | 4.0±1.9 | <0.0001 |
| Health Motivation | 12-60 | 48.0±5.5 | 47.0±6.1 | 0.061 | 51.0±3.8 | 41.0±8.1 | <0.001 | 53.0±3.1 | 47.0±5.5 | 0.135 |
| Knowledge | 0-17 | 4.0±1.8 | 4.0±2.1 | 0.072 | 13.0±1.6 | 6.0±1.8 | <0.001 | 13±1.8 | 6±1.8 | <0.0001 |
| Comprehensive knowledge | 0-17 | 9.5 | 9.5 | 0.063 | 66.3 | 11.3 | <0.001 | 66.1 | 11.3 | <0.0001 |
| Likelihood of BSE | Mean | 31.3 | 30.2 | 0.071 | 59.1 | 32.6 | <0.001 | 58.8 | 32.5 | <0.0001 |

*Based on estimated marginal means; b. Adjustment for multiple comparisons: Bonferroni.

**Table 4. General regression model analysis for repeated measure to see the effect (Mean Difference) of the intervention in outcome.**

| Variables | Times | | Mean Difference (Intervention Vs. control) | Standard Error | p-value | 95% Confidence Interval for mean difference | |
|---|---|---|---|---|---|---|---|
| | | | | | | Lower Bound | Upper Bound |
| Susceptibility score | Time 1 | Time 2 | 1.415 | 0.209 | <0.001 | 0.915 | 1.916 |
| | | Time 3 | 1.383 | 0.219 | <0.001 | 0.856 | 1.909 |
| Severity score | Time 1 | Time 2 | 7.446 | 0.377 | <0.001 | 6.541 | 8.351 |
| | | Time 3 | 7.607 | 0.380 | <0.001 | 6.696 | 8.519 |
| Benefits score | Time 1 | Time 2 | 3.600 | 0.436 | <0.001 | 2.553 | 4.647 |
| | | Time 3 | 3.505 | 0.442 | <0.001 | 2.445 | 4.565 |
| Barriers score | Time 1 | Time 2 | −3.697 | 0.494 | <0.001 | −4.883 | −2.511 |
| | | Time 3 | −3.495 | 0.510 | <0.001 | −4.717 | −2.272 |
| Self-efficacy score | Time 1 | Time 2 | 4.475 | 0.457 | <0.001 | 3.379 | 5.571 |
| | | Time 3 | 4.476 | 0.455 | <0.001 | 3.383 | 5.568 |
| Health motivation score | Time 1 | Time 2 | 1.227 | 0.318 | <0.001 | 0.464 | 1.990 |
| | | Time 3 | 2.663 | 0.260 | <0.001 | 2.039 | 3.287 |
| Cues to action score | Time 1 | Time 2 | 0.544 | 0.098 | <0.001 | 0.308 | 0.780 |
| | | Time 3 | 0.560 | 0.094 | <0.001 | 0.334 | 0.785 |
| Knowledge score | Time 1 | Time 2 | 5.545 | 0.099 | <0.001 | 5.307 | 5.783 |
| | | Time 3 | 5.529 | 0.095 | <0.001 | 5.302 | 5.756 |

*Based on estimated marginal means; b. Adjustment for multiple comparisons: Bonferroni.

### Effects intervention on each constructs (perceptions of variance explained)

Table 5 presents the effect of the community-based intervention on each construct, showing the variance explained by the intervention. The intervention accounted for 81.0% of the variance in knowledge, with a statistically significant impact on the intervention group (p<0.001). It also had a significant effect on health motivation, explaining 43.1% of the variance (p<0.001).

Regarding threat appraisal, the intervention explained 21.7% of the variance in perceived susceptibility and 24.5% in perceived severity of breast cancer, both showing statistically significant effects in the intervention group (p<0.001). However, the program explained the least variance only 1.1% in cues to action, though this effect was still statistically significant (p=0.004) (Table 5).

### Effect of interventions on likelihood of performing breast self-examination

The actual breast screening behavior was assessed as a past behavior. Table 6 presents the effect size measured for dichotomous variables on screening behavior by the intervention using nonparametric tests (Cochran Q). Accordingly, the effects of intervention were demonstrated by hearing about breast cancer and performing breast self-examination at different time points or under different conditions, which had a statistically significant effect on the study population (p<0.001). However, in the case of friends as a source of information, it indicated that the percentages or proportions at the different time points or under the different conditions are the same in the study population. Concerning the source of information, the exposure of participants to media and health worker information significantly increased after intervention and was maintained at the maintenance stage (six months) (p<0.001). However, the intervention had no significant effect on producing exposures to friends' messages (p=0.413). The frequency of breast screening had significantly increased after intervention (at least sometimes) in the last six months (p<0.001) (Table 6). However, breast screening behavior immediately after intervention had no significant effect on screening.

**Table 5. General regression model analysis for repeated measures after adjustment for multiple comparisons to see the effect of intervention on the program (dependent variable).**

| Variables | Source | Df | Mean Square | F | p-value | Partial Eta Squared |
|---|---|---|---|---|---|---|
| **Knowledge score** | Intercept | 1 | 136400.193 | 35937.169 | <0.001 | 0.979 |
| | Intervention | 1 | 12568.878 | 3311.505 | <0.001 | 0.810 |
| **Susceptibility** | Intercept | 1 | 783664.873 | 47879.105 | <0.001 | 0.984 |
| | Intervention | 1 | 3526.629 | 215.464 | <0.001 | 0.217 |
| **Severity** | Intercept | 1 | 4506599.604 | 86642.524 | <0.001 | 0.991 |
| | Intervention | 1 | 13127.782 | 252.391 | <0.001 | 0.245 |
| **Benefit** | Intercept | 1 | 7172784.850 | 109005.975 | <0.001 | 0.993 |
| | Intervention | 1 | 13064.192 | 198.539 | <0.001 | 0.204 |
| **Barrier** | Intercept | 1 | 12199491.540 | 147041.048 | <0.001 | 0.995 |
| | Intervention | 1 | 58938.209 | 710.385 | <0.001 | 0.478 |
| **Self-efficacy** | Intercept | 1 | 5709364.740 | 72790.201 | <0.001 | 0.989 |
| | Intervention | 1 | 20452.199 | 260.750 | <0.001 | 0.252 |
| **Cues to action** | Intercept | 1 | 39533.395 | 10649.094 | <0.001 | 0.932 |
| | Intervention | 1 | 31.082 | 8.372 | 0.004 | 0.011 |
| **Health motivation** | Intercept | 1 | 5339091.815 | 182773.287 | <0.001 | 0.996 |
| | Intervention | 1 | 17174.261 | 587.927 | <0.001 | 0.431 |

*Bonferroni Tests of Between-Subjects Effects of pairwise comparisons at crude level.

### The interactions/effects of HBM Constructs on likelihood of screening behavior

Table 7 presents the direct and indirect effects obtained from the path analysis model, which was used to explore the interactions among the constructs of the Health Belief Model (HBM). The path coefficients for the full model (including all directional arrows) were derived through a series of layered multiple regression analyses. In each regression, the criterion variable was the one in the designated box (excluding the leftmost layer), and the predictors were all variables with arrows pointing toward that box. Path analysis was done to determine the direct and indirect effects of variables (HBM) and to estimate the values of coefficients in the underpinning linear model. Path analysis is simply a standardized partial regression coefficient partitioning the correlation coefficients into measures of direct and indirect effects of a set of independent variables on the dependent variable. Model identification refers to determining the number of parameters to be estimated such as path coefficients and correlations relative to the amount of information available from the data, particularly the observed variances and covariance of the variables. Since our intention was to see the relationship between the variables, considering the coefficients or corresponding number of coefficients deprived showed the magnitude in which the arrows are explaining or imposing the effect on the likelihood of taking action. The greater estimation was demonstrated in perceived benefits and barriers (Table 7).

## Discussion

A cluster randomized controlled trial was conducted to evaluate the effectiveness of a community-based educational intervention on breast self-examination among reproductive-aged women in Ethiopia. The likelihood of performing breast self-examination was then assessed based on the core constructs of the Health Belief Model (HBM) [22]. Accordingly, the study found that the likelihood of performing breast self-examination and the participants' level of comprehensive knowledge were 33.3% and 11.7%, respectively.

Following the intervention (as indicated by RCT findings), the likelihood of performing breast self-examination and the comprehensive knowledge of the participants were 59.9% and 69.1% at the end of six months, respectively. In countries

**Table 6. Effect size measured for dichotomous variables on screening behavior by the intervention using non-parametric tests (Cochran Q).**

| Variables | Categories | Measurement of time (Breast screening behavior) | | | | | | | |
|---|---|---|---|---|---|---|---|---|---|
| | | Baseline | | At 3 months | | At 6 months | | Effect size | |
| | | Intervention n(%) | Control n(%) | Intervention n(%) | Control n(%) | Intervention n(%) | Control n(%) | At 3 months n(%) | At 6 months n(%) |
| Ever heard BC? | Yes | 383 (94.6) | 391 (97.0) | 393 (100.0) | 394 (98.7) | 382 (100.0) | 391 (98.7) | 22.5 (p<0.001) | 22.5 (p<0.001) |
| Heard screening methods? | Yes | 161 (39.8) | 138 (34.1) | 393 (100.0) | 142 (35.3) | 382 (100.0) | 140 (35.4) | 140.1 (p<0.001) | 132.2 (p<0.001) |
| Source of information | Health worker | 157 (97.5) | 119 (86.2) | 393 (100.0) | 124 (87.7) | 382 (100.0) | 122 (87.1) | 95.0 (p=0.025) | 95.0 (p=0.025) |
| | Media | 99 (61.5) | 84 (60.9) | 177 (45.0) | 72 (50.7) | 173 (45.3) | 71 (50.7) | 62.2 (p<0.001) | 61.7 (p<0.001) |
| | Relative | 99 (61.5) | 84 (60.9) | 139 (35.4) | 91 (64.1) | 135 (35.3) | 89 (63.6) | 75.1 (p<0.001) | 74.1 (p<0.001) |
| | Friends | 81 (50.3) | 59 (42.8) | 167 (42.5) | 68 (47.9) | 164 (42.9) | 67 (47.9) | 0.7 (p=0.413) | 0.7 (p=0.413) |
| BC screened | Yes | 47 (29.2) | 23 (17.2) | 107 (27.2) | 32 (22.5) | 99 (25.9) | 22 (15.7) | 70.0 (p<0.001) | 70.0 (p<0.001) |
| Method of screening used? | Mammography | 3 (6.4) | 3 (13.0) | 3 (2.8) | 2 (6.3) | 6 (10.6) | 5 (21.0) | 3.0 (p=0.083) | 1.0 (p=0.317) |
| | BCE | 12 (25.5) | 5 (21.7) | 9 (8.4) | 5 (15.6) | 9 (9.1) | 5 (21.7) | 3.0 (p=0.083) | 3.0 (p=0.083) |
| | BSE | 32 (68.1) | 15 (65.2) | 95 (88.8) | 25 (78.1) | 90 (90.9) | 18 (78.3) | 73.0 (p<0.001) | 61.0 (p<0.001) |
| Time of screening | 2 months Before | 5 (10.6) | 2 (8.7) | 7 (6.5) | 2 (6.3) | 7 (7.1) | 2 (8.7) | 2.0 (p=0.157) | 2.0 (p=0.157) |
| | 6 months Before | 29 (61.7) | 9 (39.1) | 64 (59.8) | 9 (28.1) | 60 (60.6) | 5 (21.7) | 35.0 (p<0.001) | 27.0 (p<0.001) |
| | A year ago | 13 (27.7) | 9 (39.1) | 36 (33.6) | 19 (59.4) | 32 (32.3) | 14 (60.9) | 33.0 (p<0.001) | 24.0 (p<0.001) |
| | 2 years ago | 0 (0.0) | 3 (13.0) | 0 (0.0) | 2 (6.3) | 0 (0.0) | 2 (8.7) | 1.0 (p=0.317) | 1.0 (p=0.317) |
| Frequency of breast screening | Sometimes | 27 (57.4) | 13 (56.5) | 65 (60.7) | 19 (59.4) | 57 (57.6) | 12 (52.2) | 44.0 (p<0.001) | 29.0 (p<0.001) |
| | Usually | 1 (2.1) | 2 (8.7) | 5 (4.7) | 2 (6.3) | 5 (5.1) | 2 (8.7) | 4.0 (p=0.046) | 4.0 (p=0.046) |
| | Consistently | 4 (8.5) | 3 (13.0) | 16 (15.0) | 7 (21.9) | 16 (16.2) | 6 (26.1) | 16.0 (p<0.001) | 15.0 (p<0.001) |
| | Others (once, ill) | 15 (31.9) | 5 (21.7) | 21 (19.6) | 4 (12.5) | 21 (21.2) | 3 (13.0) | 5.0 (p=0.025) | 4.0 (p=0.046) |

Nonparametric tests (Cochran Q).

**Table 7. Direct and indirect effects obtained by Path Analysis Model to see the interactions/effect on likelihood of taking action.**

| Path predictor Variables/Path Through | Causal Effect | | |
|---|---|---|---|
| | Direct | Indirect | Total |
| Perceived susceptibility with likelihood of breast screening | 0 | 0.02*0.54 | 0.01 |
| Perceived severity with likelihood of breast screening | 0 | 0.08*0.54 | 0.04 |
| Perceived benefits with likelihood of breast screening | 0.72*0.54 | 0 | 0.39 |
| Perceived barriers with likelihood of breast screening | 0.54*0.54 | 0 | 0.29 |
| Self-efficacy with likelihood of breast screening | 0.06*0.54 | 0 | 0.03 |
| Perceived threat with likelihood of breast screening | 0.15*0.54 | 0 | 0.08 |
| Cues to action with likelihood of breast screening | 0 | 0.04*0.54 | 0.02 |
| Health motivation with likelihood of breast screening | 0 | 0.05*0.54 | 0.03 |

*Path Analysis Model.

where the majority of women have limited formal education, such as Ethiopia, repeated educational efforts have a significant impact on reproductive-aged women at least in fostering the intention to perform breast self-examinations. The study also confirmed that community-based interventions significantly increased the likelihood of performing breast self-examinations.

In this study, the intervention increased both awareness and the actual screening practices among reproductive-aged women. However, the effect of the intervention was either maintained or slightly declined by the end of six months, indicating that ongoing education is necessary to sustain positive behavioral changes. Similarly, studies from South African have shown that the uptake of screening among women increased following educational interventions [23,24]. This is also supported by the concept of health promotion, which believes education is not a one-time affair but rather a continuous process [16,22]. Previous studies have documented that educational interventions have a considerable effect on increasing the awareness of a large audience [25–29]. The current study also revealed that the participants in the intervention arm tended to be more knowledgeable than their counterparts in the control arm, explaining the highly significant variation after intervention.

Previous studies ascertained that the perception and knowledge of breast cancer and prevention methods increased after intervention [29]. Similarly, this study found a strong relationship between perception and breast self-examination after the intervention, indicating that participants in the intervention districts were more likely to perceive themselves as susceptible. This finding is consistent with previous studies [29–31]. This is congruent with the assumption of HBM that perceived susceptibility is better than real susceptibility to facilitate preventive health behaviors [16,22]. In this study, the severity of breast cancer was found to be the best predictor of the likelihood of screening behavior after intervention, producing a highly significant mean difference in the intervention group. The result is consistent with the studies conducted in Ethiopia and other countries [3,32].

Previous studies have documented that beliefs about the cause of the disease are a crucial determinant of subsequent screening behavior [3,33]. The current study also found that the perceived benefits assuming no significant barriers of screening methods had a significant effect on promoting a preferred, positive course of action. Surprisingly, this study found that the intervention group reported higher barrier scores than the control group. Previous research outputs documented contradicting findings [28,34]. The current study also found that intervention had significantly increased women's self-efficacy in breast screening. Similarly, self-efficacy is a predicting variable in several breast cancer studies and other studies too [27,28,34]. Previously documented studies in Ethiopia have shown the influence of the media was very poor for a variety of economic, psycho-social, and cultural reasons [3,35,36]. In line with this, the current study found that the influence of cues to action following the intervention explained the least variance in the program outcomes.

In real science, health education is the best predictor of behavioral change [22]. The current study also found that practice of actual breast self-examination increased after intervention, where no cost is required. Previously published studies also supported this idea [27,29]. This study found that health motivations were increased in the intervention arm. Previous publications also confirmed that successful motivational interventions confirm the persuasiveness of individualized risks [13–15,37,38] and other intention and previously published efficacy related studies [39,40].

As a strength of the study, the enhanced control over the intervention enhances the ability to see the clear difference between the intervention and control groups. The other strength is that it might help researchers to recognize the value of interventions and explore new intervention mechanisms with an accurate method rather than mere description. The other strength of the study is that the clear yield of an estimate of the effect is unbiased on the result for intervention and control groups. As a limitation of the study, the study used health promotion interventions at the community level in case information contamination may exist due to the nature of health promotion research. Though the intervention document was reviewed by local experts and prepared for reproductive-aged women, assuming the WHO guidelines for breast self-examination but not an officially approved protocol. The exact composition and peak of health education might not be known, or insufficiency may occur in some cases.

In conclusion, the findings of this study revealed a strong interplay between the likelihood of performing breast self-examination and perceptions, knowledge, and health motivations. The study also indicated that a community-based intervention, followed by an exploratory qualitative approach through gap analysis, had a significant impact on the likelihood of performing breast self-examination. For future researchers, though RCT is the gold standard design for determining cause-and-effect relationships in its effectiveness, various longitudinal research is needed to evaluate the effectiveness of community-based educational intervention research. Breast cancer prevention and control offices should give community-based educational intervention programs to enhance women's self-efficacy in breast self-examination. In our quest to understand breast self-examination better, field specialists should figure out the intensity and range of information to determine the optimal intervention dose, intensity and peak.

## Supporting information

**S1 File. Initial study protocol.**
(DOCX)

**S2 File. SPSS data.**
(RAR)

**S1 Table. CONSORT 2010 checklist.**
(DOC)

## Acknowledgments

First, we would like to express our gratitude to the International Campus of Tehran University of Medical Sciences for the scholarship grant, as well as for the correct review and approval of this thesis work. We are also grateful to Wachemo University for duplicating questionnaires. We also want to thank the study participants for their substantial contributions to the research process.

## Author contributions

**Conceptualization:** Feleke Doyore Agide, Zewdie Birhanu.

**Data curation:** Feleke Doyore Agide, Gholamreza Garmaroudi, Roya Sadeghi, Elham Shakibazadeh, Zewdie Birhanu.

**Formal analysis:** Feleke Doyore Agide, Gholamreza Garmaroudi, Roya Sadeghi, Elham Shakibazadeh, Mehdi Yaseri, Zewdie Birhanu.

**Funding acquisition:** Feleke Doyore Agide.

**Investigation:** Feleke Doyore Agide, Roya Sadeghi, Elham Shakibazadeh, Zewdie Birhanu.

**Methodology:** Feleke Doyore Agide, Roya Sadeghi, Elham Shakibazadeh, Mehdi Yaseri, Zewdie Birhanu.

**Project administration:** Feleke Doyore Agide, Gholamreza Garmaroudi, Roya Sadeghi, Zewdie Birhanu.

**Resources:** Feleke Doyore Agide, Gholamreza Garmaroudi, Roya Sadeghi, Elham Shakibazadeh, Zewdie Birhanu.

**Software:** Feleke Doyore Agide, Gholamreza Garmaroudi, Roya Sadeghi, Mehdi Yaseri, Zewdie Birhanu.

**Supervision:** Feleke Doyore Agide, Gholamreza Garmaroudi, Roya Sadeghi, Elham Shakibazadeh, Mehdi Yaseri, Zewdie Birhanu.

**Validation:** Feleke Doyore Agide, Gholamreza Garmaroudi, Roya Sadeghi, Elham Shakibazadeh, Mehdi Yaseri, Zewdie Birhanu.

**Visualization:** Feleke Doyore Agide, Gholamreza Garmaroudi, Roya Sadeghi, Elham Shakibazadeh, Mehdi Yaseri, Zewdie Birhanu.

**Writing – original draft:** Feleke Doyore Agide.

**Writing – review & editing:** Feleke Doyore Agide.

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
