## [Decision Letter · Decision Letter 0]

9 Mar 2025

Dear Dr. Feleke Doyore Agide,

Thank you for submitting your manuscript to PLOS ONE. After careful consideration, we feel that it has merit but does not fully meet PLOS ONE’s publication criteria as it currently stands. Therefore, we invite you to submit a revised version of the manuscript that addresses the points raised during the review process.

Kindly find below the key points for revision based on the feedback:

How was the risk of unintended knowledge transfer between the intervention and control groups addressed, considering that both groups interacted with the same health extension workers?What steps were taken to minimize this exposure, ensuring that the control group received no content related to the intervention beyond what was intended?What specific measures were implemented to ensure the integrity of the intervention was maintained?Reframe conclusions to align with the study’s specific scope.Ensure findings are directly tied to data collected within the intervention context, not external studies.Proofread for grammatical errors, typos, and punctuation mistakes to improve clarity.Clearly state the Data Availability Statement in line with PLOS ONE’s Data Policy.Address all reviewer comments thoroughly, providing clear justifications.==============================Please submit your revised manuscript by Apr 23 2025 11:59PM. If you will need more time than this to complete your revisions, please reply to this message or contact the journal office at plosone@plos.org . Please include the following items when submitting your revised manuscript:

We look forward to receiving your revised manuscript.

Kind regards,

Trhas Tadesse Berhe, PhD

Academic Editor

PLOS ONE

Journal Requirements:

3. We note that the original protocol that you have uploaded as a Supporting Information file contains an institutional logo. As this logo is likely copyrighted, we ask that you please remove it from this file and upload an updated version upon resubmission.

Reviewers' comments:

Reviewer's Responses to Questions

**Comments to the Author**

1. Is the manuscript technically sound, and do the data support the conclusions?

Reviewer #1: Yes

Reviewer #2: Yes

Reviewer #3: Partly

2. Has the statistical analysis been performed appropriately and rigorously?

Reviewer #1: No

Reviewer #2: Yes

Reviewer #3: Yes

3. Have the authors made all data underlying the findings in their manuscript fully available?

Reviewer #1: Yes

Reviewer #2: Yes

Reviewer #3: No

4. Is the manuscript presented in an intelligible fashion and written in standard English?

Reviewer #1: Yes

Reviewer #2: Yes

Reviewer #3: Yes

Reviewer #1: A two-phase randomized trial was conducted which aimed to design and evaluate the effect of community-based intervention on breast self-exam among women in Ethiopia. The likelihood of both performing breast self-examination and the comprehensive knowledge increased from baseline to the end of the intervention period. Several other conclusions are unclear.

Major revisions:

1- Data collection, quality management, processing, and analysis section: The chi-square and ANOVA tests are not appropriate for comparing rates of breast self-examination at baseline, after three months and at the end of six months since an underlying assumption of these tests is independent samples.

2- Table 5: In the title indicate if these results are from univariate or multivariate models. They appear to be univariate models, meaning that only one variable at a time has been included in the model. Consider fitting a multivariate model.

3- Thoroughly explain the data presented in Table 7.

Minor revisions:

1- In the statistical analysis section, identify the statistical testing methods used to estimate the p-values in Table 1

2- P-values never equal zero; express small p-values as < 0.001.

3- Express p-values more precisely than “p< 0.05.”

4- To assist in the review process, add line numbering to the document.

Reviewer #2: I enjoyed reading this insightful and informative submission and consider it a significant contribution to the literature on women's health. However please address some minor comments to enhance the quality of the manuscript.

Abstract

Method section

It is better to write the study design method. This is a mixed-method study.

It is better to write briefly the eligibility criteria for participants, interventions intended for each group, how participants were allocated to interventions, tools used, time, and the settings where the data were collected.

Findings

State baseline characteristics at the beginning of the findings.

Introduction

The introduction was well written.

Method

What do you mean by the following phrase for Exclusion criteria?” people who could move during the intervention,”

Given that the qualitative results have been published elsewhere, it would have been better if you had not mentioned the two-phase study design and had only described the second phase.

How was the validity and reliability of the constructed instrument achieved?

Considering that the intervention content is prepared from the qualitative phase, it is better to also include a description of the content of the sessions.

The phrase "oral informed consent" is repeated twice in one paragraph.

If done, who was blinded after assignment to interventions (for example, participants, care providers, those assessing outcomes) and how

It is better to separate the data analysis from the data collection in the method section.

It would be better to delete the following sentence:” then analyzed and interpreted by the research team and a biostatistician”

Do you have any potential confounders? How to control confounding effects by statistical analysis.

Which test was used to test the normality of the distribution of the response variables?

Please write the significance level in the analysis section.

Findings

The figures are not clear.( Words are unreadable.)

In all tables, 0.000 should be corrected as follows <0.001

Please indicate the type of statistical tests used at the bottom of the tables.

What is meant by the two interventions in the following sentence?” there was no significant difference between the two interventions (6±1.75 and 6±1.78)”

How was the effect of the initial significant difference in educational and occupational status between the intervention and control groups on the outcome controlled?

Explain how missing data were addressed.

Please state the reasons for excluding from analysis.

Discussion

Considering the large volume of findings, they are not well addressed in the discussion section.

Interpretations of the results and the consistency and inconsistency of the findings with other studies need to be improved.

Reviewer #3: The manuscript's intention to see the effecct of community based edducational intervention on improving BSE practice is superb. However, the conclusion is made not only from this particular manuscript but other preceeding and related thesis works. Therefore, the conclusion should be modified based on this interventional study's findings. The authors used different stastical techniques to prove their point, but the manuscript lacks clarity on the measurement of outcome variable(s). The methods section should elaborate why they prefere each method of analysis in detail. The design itself is not clear from the title whether it is cluster randomized trial or clinical trial. The randomization process differs for clusters and individuals, so it should be clear and its seems a mixture of bothe techniques from the manuscript. There are few areas that needs correction on grammer, punctuation and typos. The data availablity is not explicitly stated in the mother document.

The comments and suggestions are given on the pdf for detail consideration, clarification and correction. Please, see the attached pdf.

I wish this paper be reconsidered by avoiding the mixups and providing clarity, so that it can help the health system to devise a startegy of primary privention for breast ca by the lower (primary) healthcare system.

**Do you want your identity to be public for this peer review?** For information about this choice, including consent withdrawal, please see our Privacy Policy

Reviewer #1: No

Reviewer #2: No

Reviewer #3: No

---

## [Author Response · Author response to Decision Letter 1]

23 Mar 2025

Response to Editors and reviewers,

We are happy to get the opportunity to revise our manuscript, titled "Designing and evaluation of the effect of community-based intervention on breast self-examination among reproductive-aged women in Ethiopia: A Cluster Randomized Trial." Manuscript number: PONE-D-23-37776. We appreciate your helpful comments, proposing an edit for further improvement, and your interest in the publication. We revised the editor's and reviewers' comments. The "Revised Manuscript" will show how much we worked on our manuscript based on your comments. The comments are valuable in improving the paper's quality. As a result, I, as the corresponding author, put the response to your comments point-by-point.

Responses to Editor

Dear respected editor, thank you for your valuable comments and interest in our manuscript for publication. Your comments were incorporated into the revised manuscript one-by-one.

Comment 1: How was the risk of unintended knowledge transfer between the intervention and control groups addressed, considering that both groups interacted with the same health extension workers?

Authors Response: Thank you for the valuable comment and professional view. We used the health extension workers from different (not included) woreda to deliver special prepared health education or intervention for intervention group; whereas, we use the existing health extension workers for the control group for usual education. We revised the manuscript in the way that your comment is being addressed.

Comment 2: What steps were taken to minimize this exposure, ensuring that the control group received no content related to the intervention beyond what was intended?

Authors Response: Thank you for the valuable professional view and comment. Since this is a health promotion intervention, we selected the intervention and control woreda from very distant places. All the intervention districts’ included kebele were categorized as intervention groups, and those control kebele were the control groups. Random assignment is made for kebele rather than individuals. Interventionists do not know for what purpose they are doing rather than recognizing the importance of intervention. We also revised this part in a way that your concern is being addressed.

Comment 3: What specific measures were implemented to ensure the integrity of the intervention was maintained?

Authors Response: Thank you for the valuable suggestion/comment. We prepared intervention protocol for the intervention delivering. i.e. Clear definition of the intervention: The intervention should be precisely defined, specifying the goals, components, target population, delivery methods, duration, and expected outcomes. A well-articulated protocol is essential for consistency across implementation. Evidence-based framework: The intervention should be grounded in a theoretical framework or an evidence-based approach that supports its feasibility and potential effectiveness in addressing the health issue based on qualitative data. Pre- and Post-Assessment: Conduct assessments before and after the intervention to measure changes in health behaviours. We supervised and trained to be always be truthful with participants, admit when they make a mistake, and treat everyone fairly. Show respect to participants and listen to their ideas. Stick to the rules and be a good role model in delivering specially prepared health education.

Comment 4: Reframe conclusions to align with the study’s specific scope.

Authors Response: Thank you for the comment. We reframed the conclusion again as per your comment to align with the study’s specific scope.

Comment 5: Ensure findings are directly tied to data collected within the intervention context, not external studies.

Authors Response: Thank you for the valuable suggestion. We revised your concern in the revised manuscript.

Comment 6: Proofread for grammatical errors, typos, and punctuation mistakes to improve clarity.

Authors Response: Thank you for the valuable suggestion/comment. We have tried to address the grammatical and punctuation issues in the revised manuscript.

Comment 7: Clearly state the Data Availability Statement in line with PLOS ONE’s Data Policy.

Authors Response: Thank you for the valuable comment. We revised Data Availability Statement accordingly.

Comment 8: Address all reviewer comments thoroughly, providing clear justifications.

Authors Response: Thank you for the valuable suggestion. We addressed all the comments of reviewers.

Responses to Reviewer #1

Thank you for your valuable comments and interest in our manuscript for publication. Your comments were incorporated into the revised manuscript and responded one by one.

General comment: A two-phase randomized trial was conducted which aimed to design and evaluate the effect of community-based intervention on breast self-exam among women in Ethiopia. The likelihood of both performing breast self-examination and the comprehensive knowledge increased from baseline to the end of the intervention period. Several other conclusions are unclear.

Authors Response: Thank you for your valuable comment. We reframed the conclusion again as per your comment to align with the study’s specific scope.

Comment 1: Data collection, quality management, processing, and analysis section: The chi-square and ANOVA tests are not appropriate for comparing rates of breast self-examination at baseline, after three months and at the end of six months since an underlying assumption of these tests is independent samples.

Authors Response: Thank you for your valuable comment. We addressed your comment in the revised manuscript.

Comment 2: Table 5: In the title indicate if these results are from univariate or multivariate models. They appear to be univariate models, meaning that only one variable at a time has been included in the model. Consider fitting a multivariate model.

Authors Response: Thank you for your valuable comment. We addressed your comment in the revised manuscript. We have also tried to clearly state how the model is fitted in the analysis part. Our analysis were done according to your comment.

Comment 3: Thoroughly explain the data presented in Table 7.

Authors Response: Thank you for your valuable comment. Path analysis was done to determine the direct and indirect effects of constructs of health belief model (HBM) and to estimate the values of coefficients in the underpinning linear model. Path analysis is simply standardized partial regression coefficient partitioning the correlation coefficients into measures of direct and indirect effects of a set of independent variables on the dependent variable. We revised your concern in the revised manuscript.

Comment 4: In the statistical analysis section, identify the statistical testing methods used to estimate the p-values in Table 1.

Authors Response: Thank you for your valuable comment. We used simply Pearson chi-square. We deleted it since it has no use.

Comment 5: P-values never equal zero; express small p-values as < 0.001.

Authors Response: Thank you for your valuable comment. We revised your concern in the revised manuscript.

Comment 6: Express p-values more precisely than “p< 0.05.”

Authors Response: Thank you for your valuable comment. We revised your concern in the revised manuscript.

Comment 7: To assist in the review process, add line numbering to the document.

Authors Response: Thank you for your valuable comment. We added the line number in the revised manuscript.

Responses to Reviewer #2

Thank you for your valuable comments and interest in our manuscript for publication. Your comments were incorporated into the revised manuscript and clarified one by one.

General comment of Reviewer #2: I enjoyed reading this insightful and informative submission and consider it a significant contribution to the literature on women's health. However please address some minor comments to enhance the quality of the manuscript.

Authors Response: Thank you for your valuable interest and comments.

Comment 1 Abstract: Method section: It is better to write the study design method. This is a mixed-method study.

Authors Response: Thank you for your valuable comment. We revised your concern in the revised manuscript.

Comment 2: It is better to write briefly the eligibility criteria for participants, interventions intended for each group, how participants were allocated to interventions, tools used, time, and the settings where the data were collected.

Authors Response: Thank you for your valuable comment. We revised your concern in the revised manuscript.

Comment 3: Findings: State baseline characteristics at the beginning of the findings.

Authors Response: Thank you for your valuable comment. We revised your concern in the revised manuscript.

Comment 4: Introduction: The introduction was well written.

Authors Response: Thank you for your encouraging comment.

Comment 5: Method: What do you mean by the following phrase for Exclusion criteria?” people who could move during the intervention,”

Authors Response: Thank you for your valuable comment. We want to say individuals who will not stay until the intervention is completed/until the last data collection time for some reasons. That means those individuals who left the environment for some reason before three times of data collection were excluded.

Comment 6: Given that the qualitative results have been published elsewhere, it would have been better if you had not mentioned the two-phase study design and had only described the second phase.

Authors Response: Thank you for your valuable comment. We revised your concern in the revised manuscript. Basically, our intention is to show that our intervention is data driven and gap recognized through qualitative method. It is a sequential exploratory design.

Comment 7: How was the validity and reliability of the constructed instrument achieved?

Authors Response: Thank you for your comment. We have conducted the Cronbanch’s alpha for reliability analysis. We have conducted pre-test in five present of the study population and the necessary adjustments were made in the questionnaire part. All this is there in the revised manuscript.

Comment 8: Considering that the intervention content is prepared from the qualitative phase, it is better to also include a description of the content of the sessions.

Authors Response: Thank you for your valuable comment. We described a description of the content of the sessions in the revised manuscript.

Comment 9: The phrase "oral informed consent" is repeated twice in one paragraph.

Authors Response: Thank you for your valuable comment. We edited in the revised manuscript.

Comment 10: It is better to separate the data analysis from the data collection in the method section. It would be better to delete the following sentence:” then analyzed and interpreted by the research team and a biostatistician”

Authors Response: Thank you for your valuable comment. We deleted and revised your concern in the revised manuscript. We also separated data analysis from the data collection in the method section.

Comment 11: Do you have any potential confounders? How to control confounding effects by statistical analysis.

Authors Response: Thank you for your valuable comment. We did regression analysis to avoid potential confounding effects and we revised your concern in the revised manuscript.

Comment 12: Which test was used to test the normality of the distribution of the response variables? Please write the significance level in the analysis section.

Authors Response: Thank you for your valuable comment. The Kolmogorov-Smirnov test was used to test the null hypothesis that a set of data comes from a normal distribution (P>0.05). We revised your concern in the revised manuscript.

Comment 13: Findings: The figures are not clear. (Words are unreadable.)

Authors Response: Thank you for your valuable comment. We revised your concern in the revised manuscript.

Comment 14: In all tables, 0.000 should be corrected as follows <0.001

Authors Response: Thank you for your valuable comment. We revised your concern in the revised manuscript.

Comment 15: Please indicate the type of statistical tests used at the bottom of the tables.

Authors Response: Thank you for your valuable comment. We revised your concern in the revised manuscript.

Comment 16: What is meant by the two interventions in the following sentence?” there was no significant difference between the two interventions (6±1.75 and 6±1.78)”

Authors Response: Thank you for your valuable comment. We edited the typos error in the revised manuscript. In the control group, the mean (4.0±2.11) showed a slight increase after three months and at the end of the intervention, whereas there was no significant difference between the two times of data collection in control group (6±1.75 and 6±1.78).

Comment 17: How was the effect of the initial significant difference in educational and occupational status between the intervention and control groups on the outcome controlled

Authors Response: Thank you for your valuable comment. Educational status is the level of education whereas occupational status is the working condition.

Comment 18: Explain how missing data were addressed. Please state the reasons for excluding from analysis.

Authors Response: Thank you for your valuable comment. To address missing data, common strategies include deletion (removing cases with missing values). We also used imputation (replacing missing values with estimates).

Comment 20: Discussion: Considering the large volume of findings, they are not well addressed in the discussion section.

Authors Response: Thank you for your valuable comment. We enriched our document and wrote the discussion part again.

Comment 21: Interpretations of the results and the consistency and inconsistency of the findings with other studies need to be improved.

Authors Response: Thank you for your valuable comment. We improved your concern in the revised manuscript.

Responses to Reviewer #3

Thank you for your valuable comments and interest in our manuscript for publication. Your comments strengthen and improved our manuscript. Thank you again. Your comments were incorporated into the revised manuscript and clarified one by one.

General Comment of Reviewer #3: The manuscript's intention to see the effect of community based educational intervention on improving BSE practice is superb. However, the conclusion is made not only from this particular manuscript but other proceeding and related thesis works. Therefore, the conclusion should be modified based on this interventional study's findings. The authors used different statistical techniques to prove their point, but the manuscript lacks clarity on the measurement of outcome variable(s). The methods section should elaborate why they prefer each method of analysis in detail. The design itself is not clear from the title whether it is cluster randomized trial or clinical trial. The randomization process differs for clusters and individuals, so it should be clear and it seems a mixture of both techniques from the manuscript. There are few areas that needs correction on grammar, punctuation and typos. The data availability is not explicitly stated in the mother document.

Authors Response: Thank you for your positive comment and professional view. We thoroughly revised your concern and the data availability is explicitly stated in the revised manuscript. General Comment of Reviewer #3: The comments and suggestions are given on the pdf for detail consideration, clarification and correction. Please, see the attached pdf.

Authors Response: Thank you for your valuable comment. We addressed comments and suggestions are given on the pdf in the revised manuscript.

Thank you again for thoroughly reviewing our manuscript!!

---

## [Decision Letter · Decision Letter 1]

29 Apr 2025

Thank you for submitting your manuscript to PLOS ONE. After careful consideration, we feel that it has merit but does not fully meet PLOS ONE’s publication criteria as it currently stands. Therefore, we invite you to submit a revised version of the manuscript that addresses the points raised during the review process.

Please complete a thorough proofread of the text and correct any spelling and grammar errors that you identify.We encourage you to revise the manuscript with careful consideration of all reviewer feedback

Indicate which changes you require for acceptance versus which changes you recommendAddress any conflicts between the reviews so that it's clear which advice the authors should followProvide specific feedback from your evaluation of the manuscript

publication criteria  and not, for example, on novelty or perceived impact.

We look forward to receiving your revised manuscript.

Kind regards,

Trhas Tadesse Berhe, PhD

Academic Editor

PLOS ONE

Journal Requirements:

Reviewers' comments:

Reviewer's Responses to Questions

**Comments to the Author**

Reviewer #1: (No Response)

Reviewer #2: All comments have been addressed

Reviewer #3: All comments have been addressed

2. Is the manuscript technically sound, and do the data support the conclusions?

Reviewer #1: Yes

Reviewer #2: Yes

Reviewer #3: Yes

3. Has the statistical analysis been performed appropriately and rigorously?

Reviewer #1: Yes

Reviewer #2: Yes

Reviewer #3: Yes

4. Have the authors made all data underlying the findings in their manuscript fully available?

Reviewer #1: Yes

Reviewer #2: Yes

Reviewer #3: Yes

5. Is the manuscript presented in an intelligible fashion and written in standard English?

Reviewer #1: Yes

Reviewer #2: Yes

Reviewer #3: Yes

Reviewer #1: Minor Revisions:

1- The standard statistical term for average is mean.

2- Table 3: Express p-values more precisely than "p<0.05".

Reviewer #2: All Comments have been addressed in the revised manuscript. Please implement these 3 comments to improve the manuscript.

Abbreviations should not be written in the abstract.( HBM)

In all tables, 0.000 should be corrected as follows <0.001. You did not write this sign[<].

Please indicate the type of statistical tests used at the bottom of the tables.

Reviewer #3: This version of the manuscript is more improved except few typos and grammatical errors. I hope the authors go over it one more time before publication.

**Do you want your identity to be public for this peer review?** For information about this choice, including consent withdrawal, please see our Privacy Policy

Reviewer #1: No

Reviewer #2: **Yes: ** Dr Somayeh Azimi

Reviewer #3: **Yes: ** Dr. Hiwot Abera Areru

---

## [Author Response · Author response to Decision Letter 2]

1 May 2025

Response to Editors and reviewers,

We are happy to get the opportunity to revise our manuscript, titled "Designing and evaluation of the effect of community-based intervention on breast self-examination among reproductive-aged women in Ethiopia: A Cluster Randomized Controlled Trial" Manuscript number: PONE-D-23-37776. We appreciate your helpful comments, proposing an edit for further improvement, and your interest in the publication. We revised the reviewer's comments. As a result, I, as the corresponding author, put the response to your comments point-by-point.

Responses to Editor

General Comments of editor: - Please complete a thorough proofread of the text and correct any spelling and grammar errors that you identify. We encourage you to revise the manuscript with careful consideration of all reviewer feedback. Indicate which changes you require for acceptance versus which changes you recommend. Address any conflicts between the reviews so that it's clear which advice the authors should follow. Provide specific feedback from your evaluation of the manuscript

Authors Response: Thank you for the valuable comment. We revised all your concerns in the revised manuscript and edited as well. We would like to inform you that we have benefited from your revision and the manuscript is improved well.

Responses to Reviewer #1

Thank you for your valuable comments and interest in our manuscript for publication. Your comments were incorporated into the revised manuscript and responded one by one.

General comment: The standard statistical term for average is mean.

Authors Response: Thank you for your valuable comment. We used mean as a statistical standard term in the revised manuscript.

General comment: Table 3: Express p-values more precisely than "p<0.05".

Authors Response: Thank you for your valuable comment. We expressed p-values more precisely in the revised manuscript.

Responses to Reviewer #2

Thank you for your valuable comments and interest in our manuscript for publication. Your comments were incorporated into the revised manuscript and clarified one by one.

General comment of Reviewer #2: All Comments have been addressed in the revised manuscript. Please implement these 3 comments to improve the manuscript. Abbreviations should not be written in the abstract (HBM), In all tables, 0.000 should be corrected as follows <0.001. You did not write this sign [<]. Please indicate the type of statistical tests used at the bottom of the tables.

Authors Response: Thank you for your valuable interest and comments. The HBM is now written in full form. We also corrected 0.000 as the <0.001 in all tables. We also indicated the type of statistical tests used at the bottom of the tables.

Responses to Reviewer #3

General comment of Reviewer #3: This version of the manuscript is more improved except few typos and grammatical errors. I hope the authors go over it one more time before publication.

Authors Response: Thank you for your valuable comments and interest. We edited for grammar and typos as much as possible.

Thank you again for thoroughly reviewing our manuscript!!

---

## [Decision Letter · Decision Letter 2]

23 May 2025

Dear Dr.Feleke Doyore Agide ,

Thank you for submitting your manuscript to PLOS ONE. After careful consideration, we feel that it has merit but does not fully meet PLOS ONE’s publication criteria as it currently stands. Therefore, we invite you to submit a revised version of the manuscript that addresses the points raised during the review process.

Ensure all reviewer comments have been fully addressedEdit for grammar, spelling, and punctuation throughout.The category “Others” in Table 2 under “Frequency of breast screening” should be clearly defined. Please specify what behaviors or responses are included in “Others.”Perform a final proofreading for flow, transitions, and logical structure.

We look forward to receiving your revised manuscript.

Kind regards,

Trhas Tadesse Berhe, PhD

Academic Editor

PLOS ONE

Journal Requirements:

Reviewers' comments:

Reviewer's Responses to Questions

**Comments to the Author**

Reviewer #1: (No Response)

2. Is the manuscript technically sound, and do the data support the conclusions?

Reviewer #1: Yes

3. Has the statistical analysis been performed appropriately and rigorously?

Reviewer #1: Yes

4. Have the authors made all data underlying the findings in their manuscript fully available?

Reviewer #1: Yes

5. Is the manuscript presented in an intelligible fashion and written in standard English?

Reviewer #1: Yes

Reviewer #1: Minor revisions:

1- Table 6: For p-values between 1.0 and 0.001, replace the less than sign with an equals sign.

**Do you want your identity to be public for this peer review?** For information about this choice, including consent withdrawal, please see our Privacy Policy

Reviewer #1: No

---

## [Author Response · Author response to Decision Letter 3]

25 May 2025

Response to Editors and reviewers,

We are pleased to get the opportunity to revise our manuscript, titled "Designing and evaluation of the effect of community-based intervention on breast self-examination among reproductive-aged women in Ethiopia: A Cluster Randomized Controlled Trial" Manuscript number: PONE-D-23-37776. We value your interest in the publication, your insightful comments, and your suggestion for an edit to make it even better. We updated the reviewers' and editor's comments. The comments are really helpful in raising the quality of the work. Since we rectified every concern brought up, we think the paper's quality has much improved. I will thus respond to each of your comments point-by-point as the corresponding author.

Responses to Editor

General Comments of editor: Ensure all reviewer comments have been fully addressed. Edit for grammar, spelling, and punctuation throughout. The category “Others” in Table 2 under “Frequency of breast screening” should be clearly defined. Please specify what behaviours or responses are included in “Others.” Perform a final proofreading for flow, transitions, and logical structure.

Authors Response: Thank you for the valuable and professional comments. We edited for grammar, spelling, punctuation, structure and flow throughout the manuscript. We specified the category “Others” and revised all your concerns in the revised manuscript and edited as well. We would like to inform you that we have benefited from your revision and the manuscript is improved well.

Responses to Reviewer #1

Thank you for your valuable comments and interest in our manuscript for publication. Your comments were incorporated into the revised manuscript and responded one by one.

General comment: 1- Table 6: For p-values between 1.0 and 0.001, replace the less than sign with an equals sign.

Authors Response: Thank you for your valuable comment. We accepted your comment and we replaced the less than sign with an equals sign in the revised manuscript. In addition to that we grammar, spelling, punctuation, structure and flow throughout the manuscript.

Thank you again for thoroughly reviewing our manuscript!!

---

## [Decision Letter · Decision Letter 3]

4 Jun 2025

Dear Dr. Agide,

Thank you for submitting your manuscript to PLOS ONE. After careful consideration, we feel that it has merit but does not fully meet PLOS ONE’s publication criteria as it currently stands. Therefore, we invite you to submit a revised version of the manuscript that addresses the points raised during the review process.

Thank you for your submission. Before we can proceed further, I kindly ask that you take the necessary time to carefully address the reviewer comments in detail. Please ensure that all required revisions are thoroughly incorporated before resubmitting the manuscript. This will help facilitate a smoother evaluation process moving forward.

We look forward to receiving your revised manuscript.

Kind regards,

Trhas Tadesse Berhe, PhD

Academic Editor

PLOS ONE

Journal Requirements:

Reviewers' comments:

Reviewer's Responses to Questions

**Comments to the Author**

Reviewer #1: (No Response)

2. Is the manuscript technically sound, and do the data support the conclusions?

Reviewer #1: Yes

3. Has the statistical analysis been performed appropriately and rigorously?

Reviewer #1: Yes

4. Have the authors made all data underlying the findings in their manuscript fully available?

Reviewer #1: Yes

5. Is the manuscript presented in an intelligible fashion and written in standard English?

Reviewer #1: Yes

Reviewer #1: Issues with the reporting of p-values persist.

1. For results where p > 0.05, please report the exact p-values to two significant digits. Additionally, for non-significant p-values, remove the less-than sign. For example, in Table 3, the p-value for Health Motivation is currently reported as '< 0.135'; this should be revised to '0.135'. Similar inconsistencies appear throughout the document and should be addressed accordingly.

2. In Table 5, change the column header from "Sig." to "p-value" for clarity and consistency.

3. When reporting p-values, always use a lowercase p. There is currently a mix of uppercase and lowercase Ps in Table 6 that should be standardized.

**Do you want your identity to be public for this peer review?** For information about this choice, including consent withdrawal, please see our Privacy Policy

Reviewer #1: No

---

## [Author Response · Author response to Decision Letter 4]

18 Jun 2025

Response to Editors and reviewers,

We are pleased to get the opportunity to revise our manuscript, titled "Designing and evaluation of the effect of community-based intervention on breast self-examination among reproductive-aged women in Ethiopia: A Cluster Randomized Controlled Trial" Manuscript number: PONE-D-23-37776. We appreciate your helpful comments, proposing an edit for further improvement, and your interest in the publication. We revised the reviewer's comments. As a result, I, as the corresponding author, put the response to your comments point-by-point.

Responses to Editor

General Comments of editor: Please review your reference list to ensure that it is complete and correct. If you have cited papers that have been retracted, please include the rationale for doing so in the manuscript text, or remove these references and replace them with relevant current references. Any changes to the reference list should be mentioned in the rebuttal letter that accompanies your revised manuscript. If you need to cite a retracted article, indicate the article’s retracted status in the References list and also include a citation and full reference for the retraction notice.

Authors Response: Thank you for the concern and professional comments. We checked references whether retracted or not.

Responses to Reviewer #1

Thank you for your valuable comments and interest in our manuscript for publication. Your comments were incorporated into the revised manuscript and responded one by one.

Comment: 1- For results where p > 0.05, please report the exact p-values to two significant digits. Additionally, for non-significant p-values, remove the less-than sign. For example, in Table 3, the p-value for Health Motivation is currently reported as '< 0.135'; this should be revised to '0.135'. Similar inconsistencies appear throughout the document and should be addressed accordingly.

Authors Response: Thank you for your valuable comment. We accepted your comment and we put the exact p-values as per your comment in the revised manuscript.

Comment: 2- In Table 5, change the column header from "Sig." to "p-value" for clarity and consistency.

Authors Response: Thank you for your valuable comment. We accepted your comment and we changed the column header from "Sig." to "p-value" in the revised manuscript.

Comment: 3- When reporting p-values, always use a lowercase p. There is currently a mix of uppercase and lowercase Ps in Table 6 that should be standardized.

Authors Response: Thank you for your valuable comment. We accepted your comment and we used a lowercase p in the revised manuscript.

Thank you again for thoroughly reviewing our manuscript!!

---

## [Decision Letter · Decision Letter 4]

14 Jul 2025

Dear Dr. Agide,

Thank you for submitting your manuscript to PLOS ONE. After careful consideration, we feel that it has merit but does not fully meet PLOS ONE’s publication criteria as it currently stands. Therefore, we invite you to submit a revised version of the manuscript that addresses the points raised during the review process.

**Please address the following formatting issue that has been raised previously but not yet fully corrected:**

**When reporting p-values that are less than 0.0001, please include the ‘<’ symbol (i.e., report as p < 0.0001). This is a standard convention to ensure precision and clarity in statistical reporting.**

We look forward to receiving your revised manuscript.

Kind regards,

Trhas Tadesse Berhe, PhD

Academic Editor

PLOS ONE

**Journal Requirements:**

Reviewers' comments:

Reviewer's Responses to Questions

**Comments to the Author**

Reviewer #1: (No Response)

2. Is the manuscript technically sound, and do the data support the conclusions?

Reviewer #1: Yes

3. Has the statistical analysis been performed appropriately and rigorously?

Reviewer #1: Yes

4. Have the authors made all data underlying the findings in their manuscript fully available?

Reviewer #1: Yes

5. Is the manuscript presented in an intelligible fashion and written in standard English?

Reviewer #1: Yes

**Reviewer #1: ** Minor revision:

If the p-values are less than 0.0001, be sure to include the '<' symbol when reporting them.

**Do you want your identity to be public for this peer review?** For information about this choice, including consent withdrawal, please see our Privacy Policy

Reviewer #1: No

---

## [Author Response · Author response to Decision Letter 5]

15 Jul 2025

Response to Editors and reviewers,

We are happy to get the opportunity to revise our manuscript, titled "Designing and evaluation of the effect of community-based intervention on breast self-examination among reproductive-aged women in Ethiopia: A Cluster Randomized Controlled Trial" Manuscript number: PONE-D-23-37776. We appreciate your helpful comments, proposing an edit for further improvement, and your interest in the publication. We revised the reviewer's comments. As a result, I, as the corresponding author, put the response to your comments point-by-point.

Responses to Editor

General Comments of editor: Please include the following items when submitting your revised manuscript:

• A rebuttal letter that responds to each point raised by the academic editor and reviewer(s). You should upload this letter as a separate file labelled 'Response to Reviewers'.

• A marked-up copy of your manuscript that highlights changes made to the original version. You should upload this as a separate file labelled 'Revised Manuscript with Track Changes'.

• An unmarked version of your revised paper without tracked changes. You should upload this as a separate file labelled 'Manuscript'.

Authors Response: Thank you for the valuable comment. We prepared and submitted 'Response to Reviewers', 'Revised Manuscript with Track Changes' and 'Unmarked version Manuscript'. We would like to inform you that we have benefited from your revision and the manuscript is improved well.

Responses to Reviewer #1

Thank you for your valuable comments and interest in our manuscript for publication. Your comments were incorporated into the revised manuscript and responded as well.

General comment: When reporting p-values that are less than 0.0001, please include the ‘<’ symbol (i.e., report as p < 0.0001). This is a standard convention to ensure precision and clarity in statistical reporting.

Authors Response: Thank you for your valuable comments and interest. We added the ‘<’ symbol before <0.0001 in all tables. I also consulted a senior statistician, he advised me to use less than sign in both cases (P<0.001 and <0.0001).

Thank you again for thoroughly reviewing our manuscript!!

---

## [Decision Letter · Decision Letter 5]

23 Jul 2025

Designing and evaluation of the effect of community-based intervention on breast self-examination among reproductive-aged women in Ethiopia: A Cluster Randomized Controlled Trial

PONE-D-23-37776R5

Dear Dr. Agide,

We’re pleased to inform you that your manuscript has been judged scientifically suitable for publication and will be formally accepted for publication once it meets all outstanding technical requirements.

Kind regards,

Trhas Tadesse Berhe, PhD

Academic Editor

PLOS ONE

Additional Editor Comments (optional):

Reviewers' comments:

Reviewer's Responses to Questions

**Comments to the Author**

Reviewer #1: All comments have been addressed

2. Is the manuscript technically sound, and do the data support the conclusions?

Reviewer #1: (No Response)

3. Has the statistical analysis been performed appropriately and rigorously?

Reviewer #1: (No Response)

4. Have the authors made all data underlying the findings in their manuscript fully available?

Reviewer #1: (No Response)

5. Is the manuscript presented in an intelligible fashion and written in standard English?

Reviewer #1: (No Response)

Reviewer #1: (No Response)

**Do you want your identity to be public for this peer review?** For information about this choice, including consent withdrawal, please see our Privacy Policy

Reviewer #1: No

---

## [Editor Report · Acceptance letter]

PONE-D-23-37776R5

PLOS ONE

Dear Dr. Agide,

I'm pleased to inform you that your manuscript has been deemed suitable for publication in PLOS ONE. Congratulations! Your manuscript is now being handed over to our production team.

Kind regards,

on behalf of

Dr. Trhas Tadesse Berhe

Academic Editor

PLOS ONE